



# Machine Learning Methods to Improve Spatial Predictions of Coastal Wind Speed Profiles and Low-Level Jets using Single-Level ERA5 Data

Christoffer Hallgren[1,*], Jeanie A. Aird[2,*], Stefan Ivanell[1], Heiner Körnich[3], Ville Vakkari[4,5], Rebecca J. Barthelmie[2], Sara C. Pryor[6], and Erik Sahlée[1]

[1]Department of Earth Sciences, Uppsala University, Uppsala, Sweden
[2]Sibley School of Mechanical and Aerospace Engineering, Cornell University, Ithaca, New York, USA
[3]Swedish Meteorological and Hydrological Institute, Norrköping, Sweden
[4]Finnish Meteorological Institute, Helsinki, Finland
[5]Atmospheric Chemistry Research Group, Chemical Resource Beneficiation, North-West University, Potchefstroom, South Africa
[6]Department of Earth and Atmospheric Sciences, Cornell University, Ithaca, New York, USA

**Correspondence:** Christoffer Hallgren (christoffer.hallgren@geo.uu.se), Jeanie A. Aird (jaa377@cornell.edu)

**Abstract.** Observations of the wind speed at heights relevant for wind power are sparse, especially offshore, but with emerging aid from advanced statistical methods, it may be possible to derive information regarding wind profiles using surface observations. In this study, two machine learning (ML) methods are developed for predictions of (1) coastal wind speed profiles and (2) low-level jets (LLJs) at three locations of high relevance to offshore wind energy deployment; the U.S. Northeastern Atlantic Coastal Zone, the North Sea, and the Baltic Sea. The ML models are trained on multiple years of lidar profiles and utilize single-level ERA5 variables as input. The models output spatial predictions of coastal wind speed profiles and LLJ occurrence. A suite of nine ERA5 variables are considered for use in the study due to their physics-based relevance in coastal wind speed profile genesis, and the possibility to observe these variables in real-time via measurements. The wind speed at 10 m a.s.l. and the surface sensible heat flux are shown to have the highest importance for both wind speed profile and LLJ predictions. Wind speed profile predictions output by the ML models exhibit similar root mean squared error (RMSE) with respect to observations as is found for ERA5 output. At typical hub heights, the ML models show lower RMSE than ERA5 indicating approximately 5% RMSE reduction. LLJ identification scores are evaluated using the Symmetric Extremal Dependence Index (SEDI). LLJ predictions from the ML models outperform predictions from ERA5, demonstrating markedly higher SEDIs. However, optimization utilizing the SEDI results in a higher number of false alarms when compared to ERA5.

## 1 Introduction

Wind energy is one of the fastest growing sources of renewable energy worldwide, representing 23% of the electricity generation from renewable sources in 2021 and approximately 136 billion kWh of electricity generated from wind energy is installed each year (EIA, 2023). Motivated by the occurrence of higher and less variable wind speeds over large bodies of water, the offshore wind energy industry is a rapidly developing and growing industry. As of 2021, 17 GW of offshore wind energy capac-





20 ity were under construction globally, with the United States and Europe implementing ambitious offshore wind energy goals
(Bojek, 2022; IRENA, 2022). The U.S. and Europe plan to install 30 and 60 GW of offshore wind energy by 2030, respectively
(American Clean Power Association, 2020; European Commission, 2020; Barthelmie et al., 2021). Along the U.S. East Coast,
lease areas for offshore wind energy projects are in various stages of development. Recent research (Pryor et al., 2021) has
suggested that the 15 northernmost lease areas could provide nearly 3% of the national electricity demand by deploying 1,922
15 MW wind turbines with a 1.85 km spacing and that offshore installments in the area are highly competitive in terms of
estimated Levelized Cost of Energy (LCoE, Foody et al. 2023). In Europe, the North and Baltic Seas are key sites for future
and current offshore wind development. In 2019, Europe's installed offshore wind energy capacity was nearly 22 GW, and the
North Sea held approximately 77% of this (Ramirez et al., 2020). In the Baltic Sea region, approximately 2.2 GW of offshore
wind power capacity was installed as of 2020, but the region is projected to produce 93 GW of offshore wind power by 2050
(COWI, 2019; Wind Europe, 2021).

Accurate measurements and predictions of the offshore wind resource are complex, especially in the coastal zone, with its
sharp transition between surfaces with different properties, i.e., roughness length and specific heat capacity (see e.g. Sempreviva
et al. 2008; Meyer and Gottschall 2022). For example, during spring and early summer, when air heated over the land surface
is advected over open water that is still cold after the winter, turbulence is suppressed and a stable marine boundary-layer can
be formed. As a result of the decrease in turbulent resistance to the air flow, the pressure gradient force becomes unbalanced,
resulting in an increase of the wind speed. This process, known as frictional decoupling, can create a local maximum in the
wind speed profile in the lowest hundreds of meters in the atmosphere (Smedman et al. 1993; Debnath et al. 2021; Hallgren
et al. 2022; also compare Luiz and Fiedler 2023). Also, differences in the specific heat capacity of land and sea surfaces
can create diurnal wind patterns known as the sea/land breeze (e.g., Miller et al. 2003; Hallgren et al. 2023b), affecting both
the vertical profiles of wind speed and wind direction. Further, coastline complexity such as capes or inlets and near-shore
topography (Burk and Thompson, 1996; Barthelmie et al., 1996; Talbot et al., 2007) affects the wind profile locally, and on the
larger scale, synoptic baroclinicity and (cold) front passages with strong vertical temperature gradients regularly cause strong
low-level flows (Kotroni and Lagouvardos, 1993; Amador, 2008). Finally, swell conditions can result in a positive momentum
flux, increasing the wind speed in the lower part of the profile, and resulting in wind profiles with negative shear at heights
relevant for wind power (Högström et al., 2009; Semedo et al., 2009; Smedman et al., 2009; Hallgren et al., 2022). Adding to
these multi-scale variabilities, atmospheric stability conditions offshore may frequently invalidate Monin-Obukhov similarity
theory assumptions (Mahrt, 1998; Newman and Klein, 2014), particularly at heights relevant for wind energy deployment,
increasing the complexity and difficulty of predicting wind speed profiles offshore.

Low-level jets (LLJs) are special cases of complex, non-ideal (i.e., non-logarithmic) wind speed profiles and are of high
relevance to the wind energy industry due to their effect on structural and aerodynamic loading, power production, and wake
recovery (Gutierrez et al., 2019; Gadde and Stevens, 2021; Gadde et al., 2021). An LLJ is generally described as a wind
speed maximum that forms within the atmospheric boundary-layer, and is associated with altered rotor plane distributions of
turbulence and shear (Pichugina et al., 2017; Aird et al., 2021). LLJs can occur offshore due to a variety of factors that manifest
from local to synoptic scales, and have been investigated offshore in numerous studies using both measurements and models



(Ranjha et al., 2013; Nunalee and Basu, 2014; Soares et al., 2020). Offshore, the frequency of LLJs varies seasonally and LLJs are most common when a stable marine boundary-layer forms under steep land-sea temperature gradients causing frictional decoupling (Angevine et al., 2006). Further, the low-level wind maximum occurs with strong gradients, having positive wind shear below the core and negative shear above the core, that possibly could prevent larger atmospheric eddies from propagating downward toward the surface — a phenomenon known as shear sheltering (Smedman et al., 2004; Prabha et al., 2008; Hallgren

et al., 2022). This introduces increases in the variance in the horizontal velocity field as vertical variance is diminished, and could possibly result in further reductions in the turbulent flux below the core.

Studies of offshore LLJs have generally indicated a peak in frequency during spring or summer, likely attributed to the advection of warm air over comparably cooler coastal waters. For example, a study of LLJs using two years of output from the Weather Research and Forecasting (WRF) model for the U.S. NE Atlantic Coast found a pronounced peak in LLJ seasonality

in June (Aird et al., 2022). Further, the study showed a significant association with low boundary-layer heights, pronounced spring/summer horizontal land-sea temperature gradients, and LLJ occurrence. A second study of an LLJ event in the coastal New York Bight Region (Colle and Novak, 2010) utilized high-resolution Light Detection And Ranging (lidar) observations and simulations from the WRF model to conclude that LLJ occurrence in the region is dependent on diurnal heating. A study of LLJs over the North Sea utilizing a combination of reanalysis data and observations from met-mast and lidar also found a

pronounced peak in LLJ frequency in the spring to early summer months (Kalverla et al., 2019). Similarly, a study utilizing four reanalyses and lidar observations over the Baltic Sea found further evidence of peak LLJ occurrence in the early spring and summer months (Hallgren et al., 2020). These findings present compelling initial evidence of unique weather conditions associated with LLJ occurrence.

As offshore wind turbine dimensions increase markedly each year – with most recent offshore wind turbine models such

as the 12 MW GE Haliade-X (General Electric, n.d.) with a hub height of 150 m and a rotor diameter of 218 m and the International Energy Agency (IEA) 15 MW offshore reference turbine (Gaertner et al., 2020) with a hub height of 150 m and a rotor diameter of 240 m – more frequent interaction between LLJs and the rotor plane is likely and LLJs have been found to occur with jet core heights and speeds relevant to wind energy over the North Sea, Baltic Sea, and U.S. Northeast Atlantic Coastal Zone (see e.g., Smedman et al. 1996; Kalverla et al. 2019; Hallgren et al. 2020; Aird et al. 2022). Further, a study of

LLJs in the North Sea (Duncan, 2018) concluded that LLJs occur with a high degree of spatial coherence – e.g., if conditions are favorable for LLJs at one location within a wind farm, the probability of LLJ occurrence throughout the entire farm may be high – which results in implications for predicting LLJ conditions across entire wind farms in real-time.

Although LLJs have been observed and simulated frequently offshore at heights relevant to wind energy, numerical weather prediction (NWP) models exhibit difficulty in resolving LLJ characteristics with high accuracy, i.e., in terms of timing and

morphology (jet core height and speed) of LLJs (Kalverla et al., 2019; Hallgren et al., 2020). Although NWP models offer higher spatial resolution than that of observational data, they are computationally expensive to run and often only cover limited geographic regions (Holt, 1996; Gevorgyan, 2018; Jiménez-Sánchez et al., 2019). Global reanalyses can provide wind speed data at sufficiently high vertical resolutions for wind power, combining past observations with modern weather forecasting models to produce meteorological variables at consistent temporal and spatial resolutions (Kaiser-Weiss et al., 2019; Hersbach



et al., 2020). However, due to limitations in computing resources or model structure, reanalysis models may not output wind speed profiles at heights relevant to wind energy or with sufficient vertical resolution for wind energy applications. While measurements in the field, i.e., from lidars, may be useful to capture wind speeds at higher vertical resolutions than those from NWP or reanalysis models, deployment can be costly and requires frequent monitoring to ensure the lidars are operating properly and gathering the desired data. Usually lidars are only available during a limited period of time and at a few selected sites. A solution to this could be to utilize continuous surface observations, at proposed or existing wind farm sites, to predict wind speed profiles.

Motivated by the previous considerations, this paper investigates the development of machine learning (ML) models to predict coastal wind speed profiles and LLJ occurrence from single-level meteorological variables. Recent studies (Vassallo et al., 2020; Bodini et al., 2023) have demonstrated promise in utilizing ML methods to vertically extrapolate wind speed profiles from lower-level measurements, as compared to traditional methods of extrapolation such as Monin-Obukhov similarity theory or the power law. Vassallo et al. (2020) applied artificial neural networks (ANNs) on a set of 11 meteorological variables and extrapolated the wind profile at three sites in different type of terrain, including a coastal site, where lidar wind profile data were available from measurement campaigns. The results were compared to standard extrapolation methods, the logarithmic wind law and the power law, and showed promising results with 65% versus 52% increase in extrapolation accuracy, respectively. Bodini et al. (2023) utilized surface-layer floating buoy observations and lidar measurements and used ML methods to extrapolate coastal wind speed profiles and quantify the long-term uncertainty of offshore model output from the WRF model in the US Atlantic Coast.

In this study, we aim to explore the feasibility and accuracy of using surface-layer measurements to predict wind speed profiles, with a focus on non-ideal wind speed profiles that may not be adequately described by Monin-Obukhov similarity theory, i.e., LLJs. The paper utilizes a combination of multiple years of the European Centre for Medium-Range Weather Forecasts (ECMWF) fifth generation reanalysis (ERA5) and lidar observations at three offshore /coastal sites in the Baltic Sea, the North Sea, and the U.S. Northeast Atlantic Coastal Zone. Two ML models, the random forest (RF) and the neural network (NN) are trained and developed to predict (1) wind speed profiles and (2) LLJ occurrence utilizing single-level variables output by ERA5. The two ML models are trained independently to verify consistency in results. The use of single-level meteorological variables for wind speed profile prediction (WSPP) and LLJ prediction is motivated by the potential utilization of observational meteorological data, e.g., wind speed or temperature measurements from meteorological masts deployed in wind farms, for real-time wind forecasting at wind farms.

The paper is structured as follows. A description of utilized data sets, the general workflow and the developed ML models are presented in Sect. 2 along with the definition used to identify LLJs and the metrics to analyze the performance of the models. In Sect. 3, the results are presented, followed by a discussion in Sect. 4. Finally, a summary of the study and some concluding remarks are provided in Sect. 5.



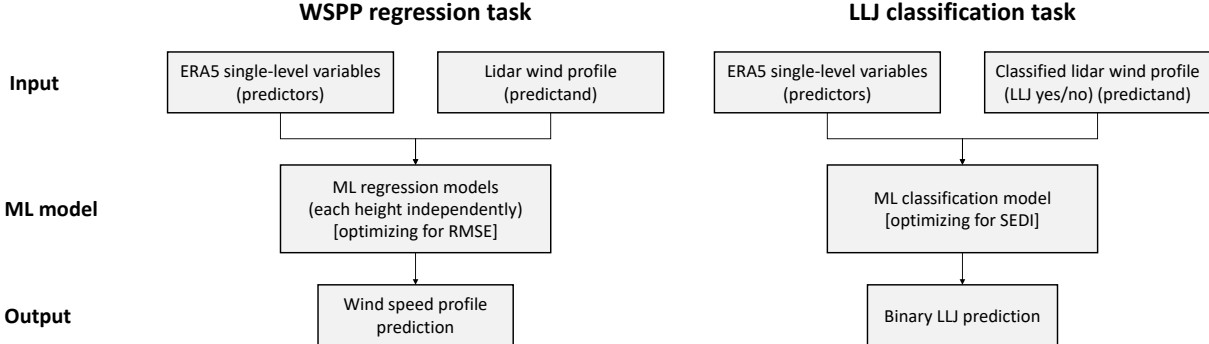

**Figure 1.** Overview of the workflow for the ML WSPP regression and LLJ classification tasks.

## 2 Materials and methods

### 2.1 Overview of workflow

Two types of supervised ML methods were investigated herein: random forest (RF) and neural networks (NNs). The models
125 were trained for each site individually, using single-level meteorological variables from ERA5 as the predictors and data from
the lidar profiles as the predictand, see workflow in Fig. 1. Both ML methods were evaluated for their fidelity in two tasks: (1)
predicting the coastal wind speed profile (regression task) and (2) predicting the LLJ (classification task) at each of the three
sites investigated in this study. Note that the predictions were performed in space, using ERA5 single-level data to predict the
speed and shape of the wind profile up to approximately 300 m a.s.l. – not in time.
130    The RF and the NN were evaluated separately and their performance for each task compared. Although the two models
differ in complexity -– with the NN being more statistically complex and requiring more computational time to train and
output predictions than the RF model -– the two models were primarily developed separately to verify that results for the
predictors with the highest importance are replicable and not just attributed to random chance or the probabilistic learning
sequence of ML models.

### 2.2 Offshore lidar measurements

The ML models were trained and tested using multiple years of lidar profiles collected at three sites in areas of high relevance
to offshore wind energy in the U.S. and Europe (Fig. 2, Table 1): the Woods Hole Oceanographic Institute (WHOI) Air
Sea Interaction Tower (ASIT) in the U.S. Northeastern Coastal Zone south of Martha's Vineyard, east Massachusetts, at the
Meteorological Mast IJmuiden (MMIJ) in the Dutch region of the North Sea, and in the Baltic Sea, on the small island of
140 Utö in the outer part of the Finnish archipelago. The sites were selected due to the availability of high quality multi-year lidar
observations and to analyse if the results hold across different offshore sites across the globe.



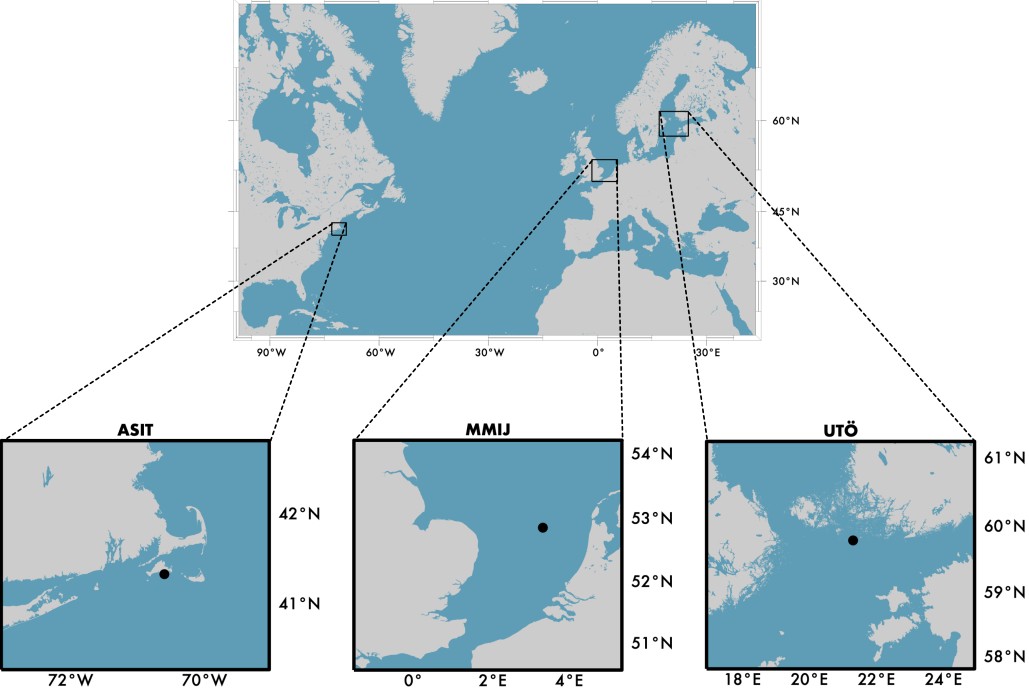

**Figure 2.** Maps of each of the three offshore sites investigated in this study -– ASIT (U.S. Northeastern Coastal Zone), MMIJ (North Sea), and Utö (Baltic Sea). Lidar locations are marked with circles.

Each lidar data set was down-sampled to hourly resolution to maintain consistency between the frequency of ERA5 output and the lidar data. Note however that data representative for a grid box from ERA5 are compare to site measurements in the study. Each site demonstrated a moderate frequency of LLJ events (defined as in Sect. 2.6, crit. 1), ranging from 2 to 10%, throughout the data collection period. In the analysis, the lidar wind speed profiles were taken as the ground truth of the actual conditions at the site and the uncertainty inherent in the measurements was not taken into account.

### 2.2.1 ASIT

The WHOI ASIT platform was installed in 2002. It is a fixed platform located 3 km south of Martha's Vineyard (Fig. 2) at a site where the water depth is 17 m (Kirincich, 2020). The ASIT is exposed to open wind and wave conditions and is located approximately 10 nautical miles from U.S. offshore wind energy lease areas near Rhode Island and Massachusetts. The wind speed measurements used herein were performed on the main platform at 13 m above sea level (a.s.l.) and were collected from a pulsed Leosphere WindCube vertically profiling lidar that was in operation Oct. 7, 2016 – Sept. 30, 2021 and then replaced by a Zephir ZX300 continuous-wave vertical scanning lidar, measuring at the same heights (51, 60, 80, 90, 100, 110, 120, 140, 160, 180 and 200 m a.s.l.). The two lidars were run in parallel Sept. 1, 2021 – Oct. 7, 2021, and were shown to be in good





**Table 1.** Lidar campaign information.

|  | ASIT | MMIJ | Utö |
|---|---|---|---|
| Latitude | 41.3332°N | 52.8482°N | 59.7791°N |
| Longitude | 70.5731°W | 3.4353°E | 21.3744°E |
| Region | East of Massachusetts | Dutch North Sea | Baltic Sea |
| Lidar Model | Leosphere WindCube V2[1] and Zephir ZX300[2] | Zephir ZX300 | Halo Photonics StreamLine |
| Temporal Resolution (min)[3] | 10 | 10 | 15 |
| Vertical Scanning Window (m) | 51–200 | 90–315 | 35–307[4] |
| Vertical Resolution (m) | 15 (averaged) | 25 | 8 |
| Collection Period | Oct. 2016 – Dec. 2022 | Nov. 2011 — Mar. 2016 | Feb. 2015 — Dec. 2022 |
| Sample Size (h)[5] | 41,353 | 36,179 | 56,257 |
| LLJ Sample Size (h)[6] | 765 | 585 | 5,528 |

[1] Oct. 7, 2016 – Sept. 30, 2021, [2] Oct 1st, 2021 – Dec 31st, 2021, [3] before downsampling, [4] as utilized in this study, [5] hours with wind speed profiles, [6] hours with an LLJ, crit. 1

agreement, assuring consistency in the data set. Previous analyses of these have indicated a high prevalence of low turbulence conditions (Bodini et al., 2020).

### 2.2.2 MMIJ

The Meteomast IJmuiden (MMIJ, Maureira Poveda and Wouters 2015) is an offshore measurement platform located approximately 85 km off of the Dutch Coast, where the water depth is approximately 28 m. Wind speed measurements utilized herein were collected from a Zephir ZX300 continuous-wave vertically profiling lidar that was deployed November 2011 — March 2016. Post-processing was conducted by the Energy Centre of the Netherlands and wind speed data are provided as 10-minute averages (Werkhoven and Verhoef, 2012). Lidar measurements have been validated, having a mean bias of 1%, using data from the cup anemometers located on the mast at 90 m. Data from the MMIJ wind lidar have been used extensively in the analysis of the North Sea LLJ, and for an in-depth analysis of temporal occurrence and LLJ morphology we refer to Kalverla et al. (2017, 2019, 2020). Measurement heights of the MMIJ lidar are 90, 115, 140, 165, 190, 215, 240, 265, 290 and 315 m a.s.l.

### 2.2.3 Utö

The Utö island is located at the southern edge of Finnish archipelago in the Baltic Sea, approximately 60 km off of the Finnish mainland. The mean depth of the Archipelago Sea north of Utö is 19 m, to the south the Baltic Proper is deeper. The Utö Atmospheric and Marine Research Station (Hirsikko et al., 2014; Laakso et al., 2018) hosts a number of measurements including a Halo Photonics Stream Line scanning Doppler lidar. The Halo lidar laser and amplifier were upgraded to the XR version in October 2017. Here, we utilised horizontal winds retrieved from 15° elevation angle conical, i.e., vertical azimuth





display, (VAD) scans at every 15 min. A threshold of the signal-to-noise ratio of 0.005 was applied to radial measurements post-processed according to Vakkari et al. (2019) before wind retrieval following Browning and Wexler (1968); the retrieved wind speed agrees well with local anemometer (Tuononen et al., 2017). Range resolution of the lidar is 30 m and the three

lowest range gates were discarded due to effects by the outgoing pulse. With the lidar located at 8 m a.s.l., this results in 7.8 m vertical resolution from 35 m a.s.l. up.

### 2.2.4   Quality control

Quality control was performed by all data providers. In addition to that, inspection of lidar profiles to exclude those with more than 75% of data points missing within a vertical profile and sizeable gaps in the profile data (only wind speed profiles with at

least five consecutive data points in a row are included in the analysis). A few profiles were removed after manual inspection, identifying malfunction of the device.

### 2.3   ERA5

The ECMWF ERA5 reanalysis is a source for global atmosphere, land surface, and ocean wave conditions from 1940 onward (Hersbach et al., 2020). ERA5 reanalysis uses the Integrated Forecasting System (IFS) Cy41r2 and replaces the former re-

analysis from ECMWF, the ERA-Interim. The ERA5 horizontal resolution is $0.25° \times 025°$, and meteorological variables are output hourly. In this study, only data from the grid point closest to the location of the lidar were used, resulting in distances of 10 km for ASIT, 12 km for MMIJ, and 7.7 km for Utö. While only single-level variables from ERA5 are used for the ML models, the ERA5 wind profile was also assessed using the wind components on terrain-following hybrid-sigma model levels. To calculate the height of the these levels, the surface pressure as well as temperature and humidity data from the model levels

were downloaded.

ERA5 is utilized widely for wind energy applications due to its consistency across the globe and has been extensively validated for use in wind energy contexts (e.g., Olauson 2018; Hallgren et al. 2020; Soares et al. 2020; Hayes et al. 2021; Pryor and Barthelmie 2021; Gualtieri 2022). ERA5 temperature, wind, and humidity profiles have been validated through comparison with radiosonde data, and ocean wave height has been validated through comparison with buoy wave data (Soares et al., 2020).

Based on the long list of single-level variables provided by ERA5, a selection of nine variables with a Pearson correlation coefficient < 0.5 was performed. The selection was based on the plausible relevance of the variable for LLJ occurrence or coastal wind speed profile forecasting (thus, the variables are physics-based) and the potential for measurability in the field, i.e., the potential to use the predictors in dynamic real-time wind speed profile prediction. The variables are the 10 m wind speed (ws10), the 10 m wind direction (wdir10), the sea surface temperature (SST), the mean sea level pressure (MSLP), the

total precipitation (precip.), the convective available potential energy (CAPE), the surface sensible heat flux (SHF), the surface net radiation (Rn), and the low cloud cover (LCC). Although difficult to measure using only surface observations, CAPE was one of the variables included, to allow for an analysis of if the available energy in the atmosphere aloft has a strong impact on the wind speed profile. The variables and the physics-based motivation for possible importance for wind profile prediction are presented in Table S1 in the Supplement.



To allow for a comparison of the ERA5 wind speed profile to the lidar wind profile, the ERA5 wind speed profile was interpolated to the same heights as in the lidar profiles. The interpolation was performed by fitting a piece-wise cubic Hermite interpolating polynomial (PCHIP) on a logarithmic height scale (Fritsch and Carlson 1980; Brodlie and Butt 1991; see also Hallgren et al. 2020).

## 2.4  Splitting the data

For each site, the time series of lidar observations and isochronal ERA5 data were split into training (approximately 60% of the data), validation (approximately 15%) and testing (approximately 15%). Temporal auto-correlation between both training and validation, and training and testing time periods was minimized through grouping the testing and validation data sets into one-week blocks that are randomly placed in the time series, and excluding one day before and after the blocks (resulting in 90% coverage). In Fig. 3, the split of the data set is shown for Utö (see supplement for ASIT and MMIJ). LLJ seasonality was

evaluated at each site for the training, validation, and testing subsets, and was well represented across each data set, i.e., both validation and testing subsets indicate the same seasonal pattern as in the training set. Although similarities between different height levels and across sites in which are the important predictors are to be expected, all models were trained independent of each other, i.e., no information was exchanged between the models.

## 2.5  ML methods

As described previously, two separate ML tasks are evaluated herein: (1) wind speed predictions at each vertical level in the observational data and (2) binary predictions of LLJs. The wind speed profile prediction (WSPP) task is a regression problem and was optimized through minimization of the root mean squared error (RMSE) with respect to the wind speed as measured by the lidars, whereas the task for binary LLJ prediction is a classification problem and was optimized through minimization of the Symmetric Extremal Dependence Index (SEDI), with the lidar observations being used as ground truth for LLJ occurrence.

In general, for both ML methods and for both tasks, the training-validation process consists of finding the optimal set of predictors. This is done by comparing the results from the forward selection and backward elimination feature selection processes (e.g., Mao 2004). After finding the optimal set of predictors, the contribution of each predictor in order to get the best score was calculated. The predictor importance, $PI$, was calculated for each predictor by evaluating which score, $\hat{s}$, that the ML model would result in for the validation period, higher RMSE or lower SEDI, if omitting the predictor from the set of

optimal predictors. The $PI$ was then calculated as

$$PI = \left| 1 - \frac{\hat{s}}{S} \right| \qquad (1)$$

where $S$ is the score for the validation period if all predictors in the optimal set are included. Important predictors result in high values of $PI$, while predictors of low importance yield low values of $PI$. As the final step, the prediction for the test period was calculated, using the optimal set of predictors.



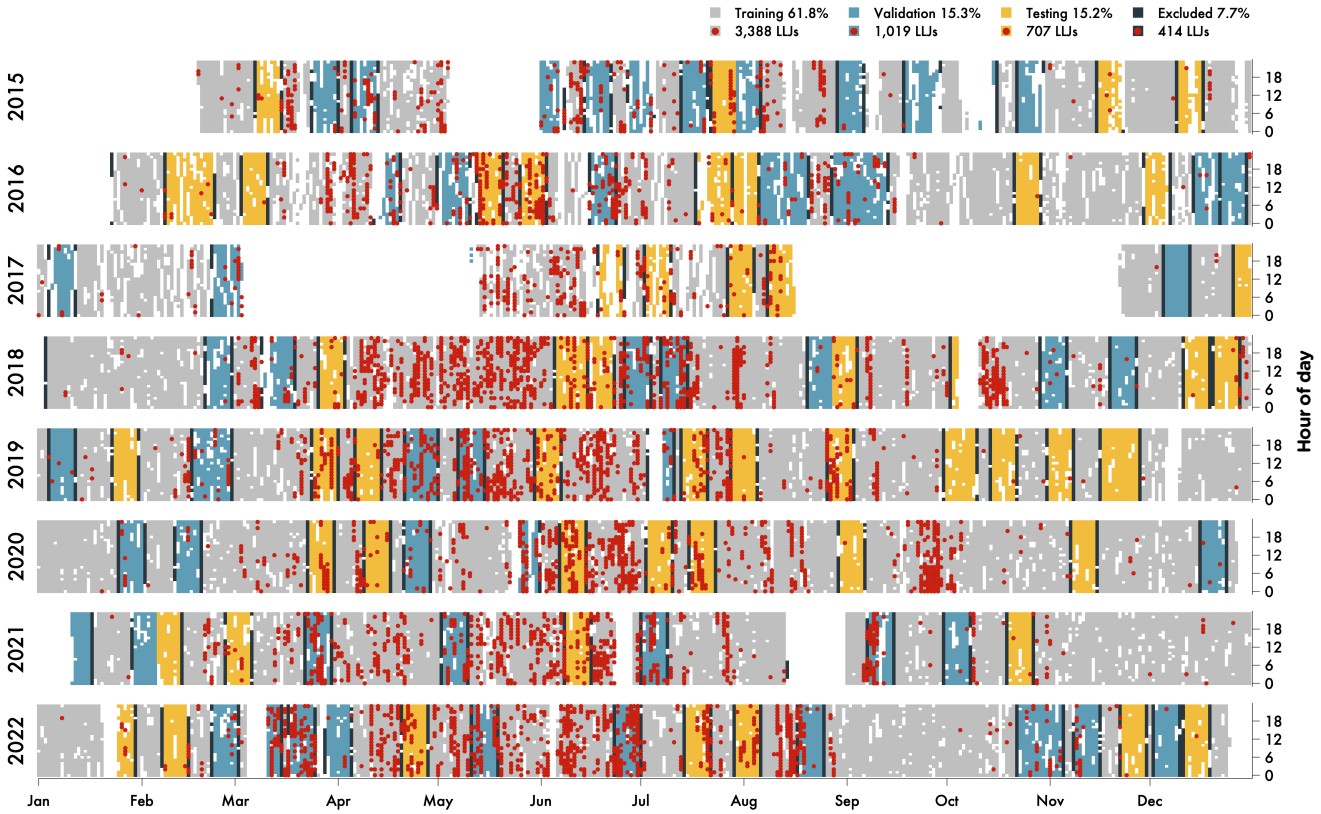

**Figure 3.** Overview of the data split for the Utö time series, Feb. 2015 – Dec. 2022. The data were split into training (light gray), validation (blue), and testing (yellow) subsets and to minimize temporal auto-correlation one day of data before and after blocks of validation and testing are excluded (dark gray). Hours where there was an LLJ in the lidar profile are marked with red circles.

### 2.5.1 Random Forest (RF)

The RF (Breiman, 2001) ML method builds on the output of a large number of individual decision trees. Each tree was fed with a random subset of the training data which minimizes the problem of overfitting, and – since each split in the tree is associated with some randomness – an individual prediction of the wind speed or the shape of the profile, LLJ yes or no, is output by each tree. These individual predictions by the trees are then merged to create a total prediction for the forest as a whole.

The number of trees in the RF is both associated with the performance and also the computational time, with more trees generally resulting in higher performance and longer computational time. In this study, 100 trees were used in forward selection and backward elimination, but increased to 500 trees for calculation of $PI$ and for the prediction for the test period. The minimum leaf size, a RF hyperparameter which is inversely proportional to the number of splits in a tree, referring to the maximum amount of data points on the final nodes in a tree, was set to the recommended values; 5 for the WSPP regression task and 1 for the LLJ classification task.





For the classification task, a cost matrix was included in the training of the RF. The cost matrix attributes a higher cost to failing to predict an LLJ compared to when falsely predicting an LLJ when there was actually no LLJ according to the lidar profile. As the cost matrix was seen to have a major impact on the performance of the RF, it is crucial to find a good cost matrix prior to the feature selection. Thus, 15 pre-defined costs matrices were tested with cost factors ranging from 1 to 30,000 and

evaluated for the validation period using all nine ERA5 variables as predictors. The cost factor resulting in the highest SEDI was then kept for feature selection and in the prediction for the test period.

### 2.5.2  Neural Networks (NN)

Two fully connected neural network architectures are implemented for the WSPP regression (using *fitnet* as included in Matlab 2021b) and LLJ classification (Vahe Tshitoyan, 2023) tasks. We refer to references therein for detailed information regarding

the network architecture.

Both NN models were optimized separately for each task due to variations in their network architecture. In the case of the classification task, the NNs were optimized for each site by finding the combination of parameters that maximize the SEDI within the optimization data subset: number and configuration of hidden layers, training iterations, and the regularisation parameter $\lambda$. For the WSPP task, the default activation function – the rectified linear unit function (for further details on

rectifiers, see He et al. 2015) – was implemented during training, and the predictors were standardized using the subset mean and standard deviation. The model was optimized via finding the number of training iterations and relative gradient tolerance for the gradient of the loss function that minimizes the RMSE. Similarly to the RF model training for the regression task, the NN was optimized using the optimization subset for each vertical level in the observational data, and both NNs (regression and classification) were optimized for each site independently.

### 2.6  Definition of the LLJ and design of preliminary sensitivity study

The LLJ definition describes which wind speed profiles are classified as LLJs and which are not. Generally, there is a high dispersion of LLJ definitions across studies, and although a shear-based definition is recommended for wind energy purposes (Hallgren et al., 2023a), an absolute and a relative criterion relating to the falloff (the decrease in wind speed) above and below the local maximum in the wind speed profile, i.e., the jet core, was applied in this study to simplify comparisons with other

work performed in the three areas (see e.g., Hallgren et al. 2020; Kalverla et al. 2020; Aird et al. 2022). As was shown by Aird et al. (2020), stronger LLJs may be linked to different causal mechanisms than weaker LLJs, and the LLJ definitions consequentially affect the core speed magnitude of wind speed profiles identified as LLJs.

A sensitivity study was performed to evaluate the significance of the ERA5 single-level variables during LLJ formation. The sensitivity study employed three LLJ criteria with increasing levels of strictness

– crit. 1: 1 m s$^{-1}$ and 10% decrease in wind speed above and below the jet core

   – crit. 2: 2 m s$^{-1}$ and 20% decrease in wind speed above and below the jet core

   – crit. 3: 3 m s$^{-1}$ and 30% decrease in wind speed above and below the jet core





and evaluates whether the changes in strictness significantly affect the distributions of the ERA5 variables during LLJ and non-LLJ hours. More specifically, the sensitivity study evaluated: (1) whether monthly median meteorological conditions during
LLJ formation are significantly different than hours without LLJs, and (2) whether monthly median meteorological conditions during LLJ occurrence are significantly different for stronger LLJs than for weaker LLJs. Monthly statistics was used based to address the large seasonal variation in LLJ frequency, see Fig. 4. These results were then used to motivate the LLJ definition utilized for both ERA5 data and lidar profiles throughout this study. For each of the three sites, Mann-Whitney U-Tests were applied to the full time series distributions of single-level meteorological variables during LLJ and non-LLJ hours. The null
hypothesis that the distributions of single-level meteorological variables from the LLJ and non-LLJ populations have the same median value was then tested, using a significance level of 0.05.

### 2.7 Wind speed profile prediction (WSPP)

To predict the wind speed in the profile, an ML model was independently trained for each level of lidar measurements, i.e., in the case of Utö, 36 independent RF and 36 independent NN models were set up. The RMSE was utilized to optimize the ML
models during training and and was calculated as

$$RMSE = \sqrt{\sum_{i=1}^{n} \frac{(\hat{u}_i - u_i)^2}{n}} \tag{2}$$

where $\hat{u}_i - u_i$ represents the residual, i.e., the difference between the predicted wind speed, $u_i$, and the lidar wind speed at the same height, $\hat{u}_i$, at time step $i$, and where $n$ is the number of samples.

### 2.8 LLJ classification task

In the classification task of predicting the LLJ, one RF model and one NN model was generated for each site. The models are optimised for the SEDI, using the LLJ occurrence as given by the lidar observations as the ground truth. The SEDI is a base-rate independent score which is non-trivial to hedge, and due to its independence of the frequency of an event, the SEDI is useful for evaluating predictions of rare events (Ferro and Stephenson, 2011). These types of events are difficult to evaluate due to the degeneration of traditional forecasting scores as events increase in rarity. The SEDI is calculated using a $2 \times 2$ contingency
table consisting of frequency of correct rejections ($cr$), false alarms ($fa$), hits ($h$), and misses ($m$) in the predictions. Using the hit rate ($H$)

$$H = \frac{h}{h+m} \tag{3}$$

and the false-alarm rate

$$F = \frac{fa}{cr+fa} \tag{4}$$

the SEDI can be calculated as

$$SEDI = \frac{logF - logH - log(1-F) + log(1-H)}{logF + logH + log(1-F) + log(1-H)} \tag{5}$$

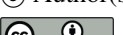



A SEDI of 1 indicates perfect prediction skill, while a SEDI of 0 indicates no more skill than random chance following the climatology. A negative SEDI indicates that the prediction is worse than what would have been expected from a climatological prediction.

## 3 Results

### 3.1 Sensitivity study of LLJ predictor significance

Results for the sensitivity study of the significance of single-level meteorological variables during LLJ occurrence are presented in Fig. 4. The three LLJ definitions, crit. 1–3, are evaluated across each site, and Mann-Whitney U tests are conducted to evaluate whether median meteorological conditions during LLJ occurrence are significantly different than those during non-LLJ hours. These statistical tests are conducted to further motivate the choice of which ERA5 single-level variables to be utilized in this study and the selection of which LLJ definition to use for training and testing the ML models. Note that this test is not applicable for wdir10, as the coordinates for that variable are circular. For all three sites, circles are plotted only if the sample size is greater than 30 LLJs in that month, resulting in only six months plotted for MMIJ, pertaining to warmer months and crit. 1, and the circles are colored if the meteorological conditions, i.e., the single-level ERA5 variables, have significantly different monthly medians during LLJ occurrence compared to hours without LLJs. All single-level ERA5 meteorological variables tested have significantly different medians during LLJ hours when compared to non-LLJ hours in at least one month at each site. Each of the three LLJ criteria with increasing strictness were evaluated, and only Utö had sufficiently high LLJ sample sizes to include results for all three criteria, due to a relative higher frequency of LLJs at Utö and a larger data set in general when compared to the other two sites. Increasing the LLJ criteria strictness results in reductions of relative LLJ frequency, with more than a 50% reduction in LLJs meeting crit. 2 as compared to crit. 1.

The highest LLJ frequency is observed in the warmer months across all three sites, with peaks at approximately 4% of hours in June for MMIJ and ASIT (crit. 1), and at approximately 25% of the hours in May for Utö (crit. 1). For all months, and all sites, the SHF demonstrates a significantly higher ensemble median values during LLJ occurrence when compared to hours without LLJs (Fig. 4, denoted with the + signs under the monthly circles at each site). Notably, the low cloud cover (LCC) is usually at lower values when LLJs are occurring, indicating a lower amount of low clouds in LLJ conditions. No clear trend is observed for the surface net radiation (Rn), while the CAPE is usually lower when LLJs occur, indicating that LLJs more frequently occur during less convective conditions. Further, conditions are usually dry and the MSLP is generally higher during LLJ occurrence. Finally, LLJs tend to appear at higher SST values than the monthly median, and the ws10 is generally lower during LLJ occurrence.

Given the evidence for significantly different meteorological conditions during LLJ occurrence for all variables considered at all sites in at least one month, the choice of ERA5 single-level variables for model training and development is validated. Further, given more consistent results and the larger sample sizes associated with LLJs extracted using crit. 1, the least strict criterion, this criterion is selected and is implemented for LLJ identification and model training, validation and testing in the following sections.

**Figure 4.** Monthly Mann-Whitney significance test results for ASIT, MMIJ, and Utö. Three LLJ criteria (see Sect. 2.6) are evaluated with increasing levels of strictness. Circles denote months with at least 30 LLJ samples as extracted with each criterion and are scaled by the monthly LLJ relative frequency for that site, also shown in the bottom panels. Colored circles indicate that median values of the single-level ERA5 variables are significantly different during LLJ and non-LLJ hours, i.e., the Mann-Whitney U Test null hypothesis is rejected at the 0.05 significance level. Plus and minus signs denote whether the medians of the variables for the LLJ samples are significantly lower or higher than for the non-LLJ samples.





## 3.2 Results for WSPP


Following the validation of the choice of ERA5 single-level variables and the LLJ definition, the ML models are developed and the ERA5 single-level variables are further examined for predictor importance in WSPP, see Sect. 2.7. The predictor importance for the validation period is calculated for each vertical level and the highest values of $PI$ are represented with colored circles in Fig. 5. For all sites, both ML methods show that the predictor with the highest importance is, as expected, ws10 followed

by SHF. For each site, these results are consistent across nearly all vertical levels (50 – 300 m a.s.l.). However, for ASIT the NN displays more dispersion identifying the important variables than the RF. The consistency in predictor importance, ws10 and SHF, across both RF and NN indicates replicability of the model development and also assures the validity of the physical importance of the predictors. The physical implications of these results are discussed in further detail in Sect. 4.

For each height level, RMSE values considering the lidar observations as the ground truth, are calculated for the test period,

both for the ERA5 profile and the resulting profiles from the ML models, see Fig. 6a–c. Extracting only cases with LLJ profiles present in the lidar profile, the RMSE of ERA5 and the relative RMSE reduction of the ML models are presented in Fig. 9d–f.

Although the WSPP is set up for height-by-height wind speed prediction not taking LLJs specifically into account, the ML models perform better than ERA5 when an LLJ is present in the lidar profile. This is evident for both the RF and the NN at heights of approximately 50–150 m a.s.l., in which the RMSE between the WSPP and the lidar profiles is approximately

10 (NN) to 20 (RF) percent lower than that of ERA5. Considering all time steps (panels a–c), the ML models demonstrate minor reduction in RMSE over ERA5, with both the RF and the NN yielding an approximate 0–10% reduction in RMSE at heights up to 150 m a.s.l. The greatest enhancement relative to the ERA5 direct model output is at levels closest to the surface; a consequence of using surface variables from ERA5 as input for the ML models. The RF model performs consistently for RMSE reduction in the ASIT test period, with an approximate 5–10% reduction in RMSE at all heights (50–200 m), when

compared to ERA5. For ASIT and MMIJ, the RF gives better results, lower RMSE, than NN across all heights. However, for Utö, the RF and the NN exhibit similar dispersion in RMSE reductions, which may indicate that the NN performance is affected by a lack of sufficient training data at the other two sites, 27–36% less training data. These discrepancies are discussed further in Sect. 4. In general, both the RF and the NN result in favorable RMSE reductions at heights relevant to wind energy, approximately 50–200 m a.s.l., when compared to the interpolated ERA5 wind speed profiles.

As the ML WSPP task generates wind speed profiles, these can be assessed for LLJs using crit 1. Morphology of the LLJs identified in the ML profiles are calculated and compared with those from the lidar and interpolated ERA5 profiles (Fig. 7). The medians, denoted with circles in Fig. 7, of the jet core heights and speed distributions show a lower dispersion for the core speeds than the core heights for all three sites, indicating that core speed distribution is easier to predict than core height distribution. For ASIT and MMIJ, ERA5 did not produce enough LLJs in the test period to generate a representative

distribution. For all three sites, both the RF and the NN generally predict higher LLJ core heights than those given in the lidar profiles. The NN generally outputs a bimodal distribution for predicted LLJ heights, particularly for ASIT and MMIJ (Fig. 7ab). This may indicate a threshold effect, that the NN associates certain predictor values with lower or higher LLJ core heights. In comparison, distributions of LLJ core speed as predicted by the NN more closely agree with the lidar observations



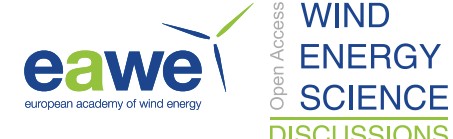

**Figure 5.** Predictor importance by height of each variable selected by the ML methods to predict the wind speed in the profile for each of the three sites and the two ML methods, top: RF, bottom: NN.The predictor importance denotes how important the predictor is to minimize RMSE between the model predictions and the ground truth lidar observations. ERA5 variables with highest predictor importance are indicated with colored circles. The predictor importance is calculated for the validation period.





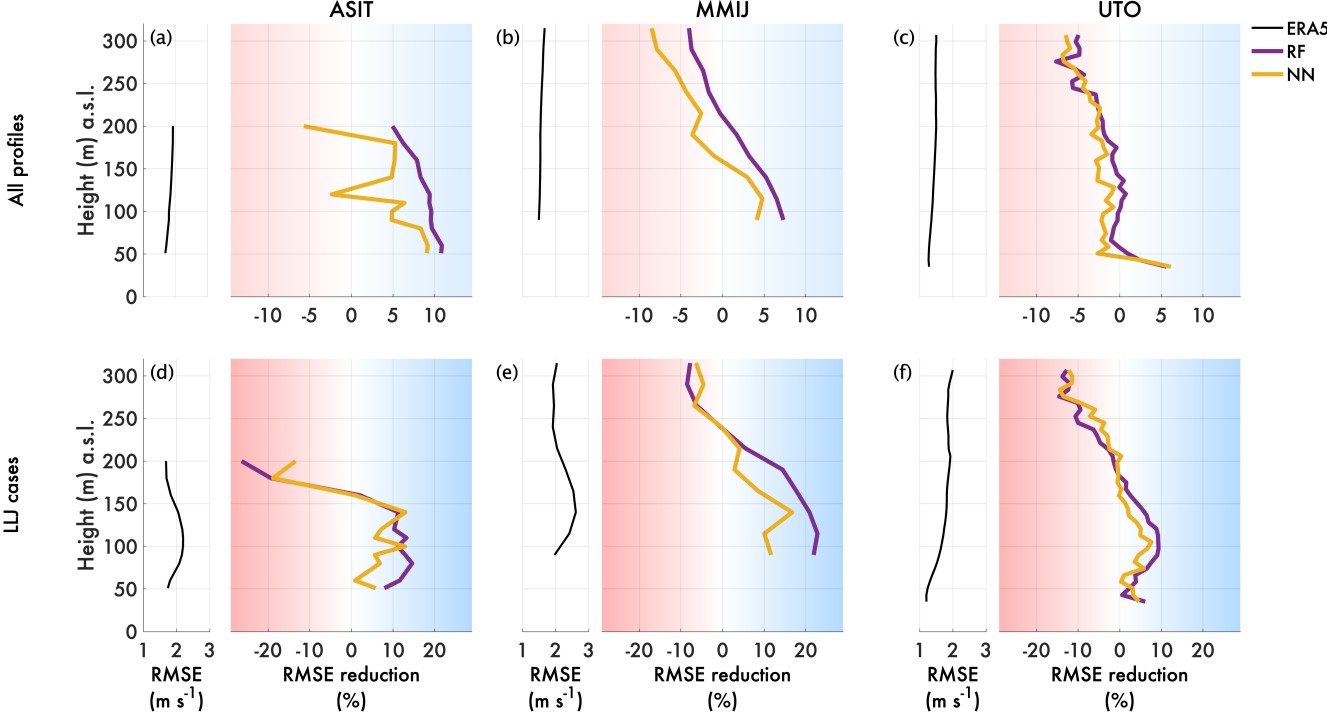

**Figure 6.** (a–c) RMSE for wind speed profiles for ERA5 (dotted) and RMSE reduction for the RF (purple) and the NN (yellow) when compared to ERA5 RMSE. The values of RMSE are calculated relative to the lidar observations and the results are valid for the test period. If the ML predicted wind speed profiles exhibit lower RMSE than ERA5, i.e., the ML methods give better results than ERA5, the RMSE reduction is positive (blue region), while worse results than ERA5 are described by negative RMSE reduction values (red region). Panels (d–f), present ERA5 RMSE and RMSE reduction of the ML models for the subset of the test data with LLJs (crit. 1) in the lidar profiles. Note the different scales on the abscissa between panels a–c and d–f.

when compared to both the RF model and ERA5. The task of LLJ core height prediction thus appears to be more difficult than the task of LLJ core speed prediction. It is possible that other surface-level variables may be more conducive to LLJ height prediction than the variables used herein, and future investigation is warranted.

### 3.3 Binary LLJ prediction

Predictor importance for the LLJ classification task during the test period is assessed, and results are presented below in Fig. 8. Here, results are only provided for the RF, as the NN becomes unstable when some predictors are missing. Higher predictor importance is associated with a greater contribution to the maximization of the SEDI. Similarly to predictor importance results for the regression task, predictors with the highest importance for wind speed profile LLJ classification are SHF and ws10. The physical significance of these predictors is discussed in further detail in Sect. 4. Testing the different LLJ criterion, crit. 1 to





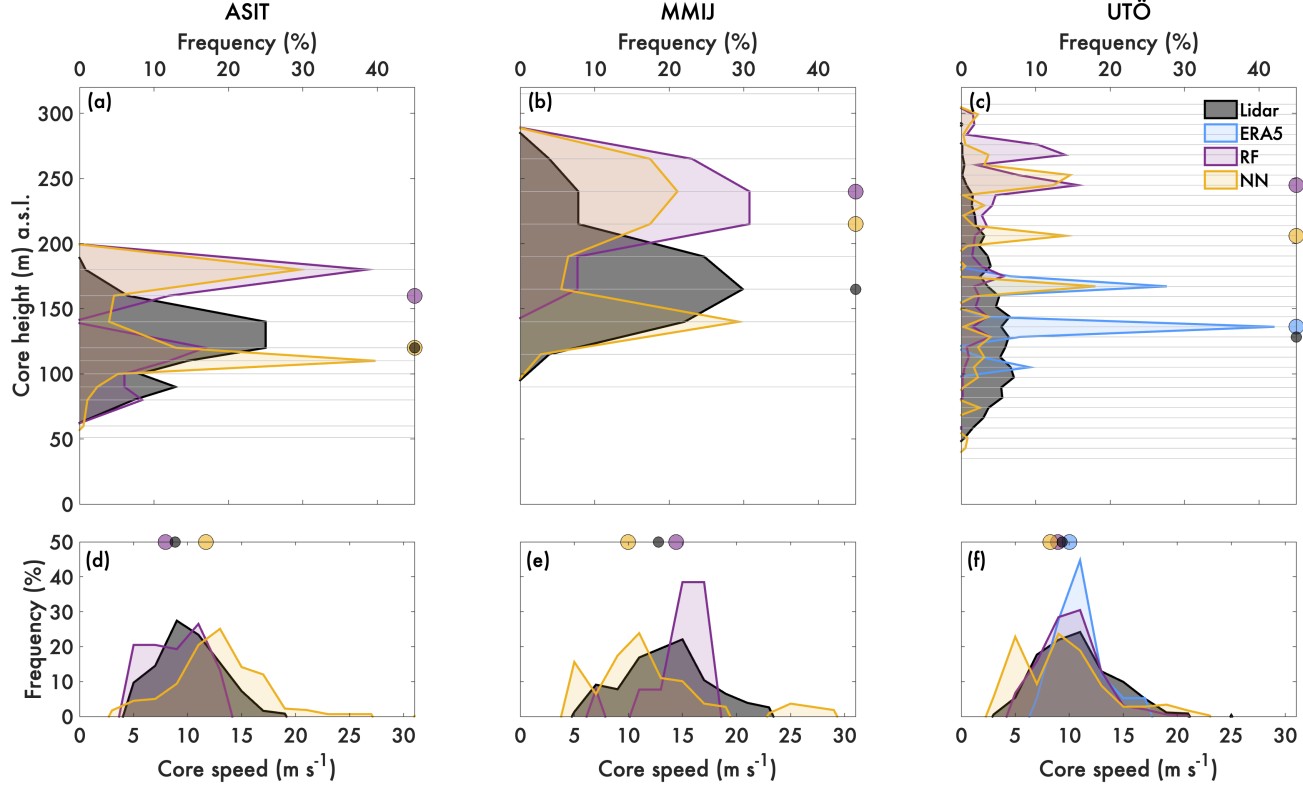

**Figure 7.** Distributions of LLJ core height (a–c) and core speed (d–f) for the three sites for the test period. For LLJ identification, crit. 1 is applied. The horizontal lines in panels a–c mark the height levels in the lidar measurements, i.e., the possible heights of LLJ cores. The circles mark the medians of each distribution, respectively. A threshold of at least 10 LLJs is set to plot the statistics, and thus, ERA5 data is missing for ASIT and MMIJ.

crit. 3, for Utö (the site with highest sample size of LLJs), the predictor importance remains consistent, with SHF and ws10 having the highest predictor importance (not shown).

In Fig. 9, SEDIs for LLJ prediction in the test period are presented for each site. Included in the figure are also LLJ SEDIs for the interpolated ERA5 wind speed profile and for profiles generated in the WSPP regression task. Generally, the RF perform better than the NN, with the RF models resulting in SEDIs of approximately 0.7 for all sites. Similar to the regression task, the NN model appear to suffer from a lack of training data, with NN results most closely matching RF results for Utö (SEDI for RF: 0.72; SEDI for NN: 0.65). However, both models perform considerably better in LLJ identification than the ERA5 wind
profiles. For both the RF and the NN, the classification models (diamonds in Fig. 9) perform notably better than the regression models (squares) for LLJ prediction. These results indicate that the classification task adds marked accuracy to predicting LLJ conditions when compared to the regression task or ERA5 alone. It is however possible that combinations of the two tasks may be explored for even further improvement.

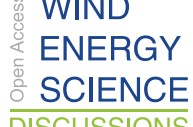

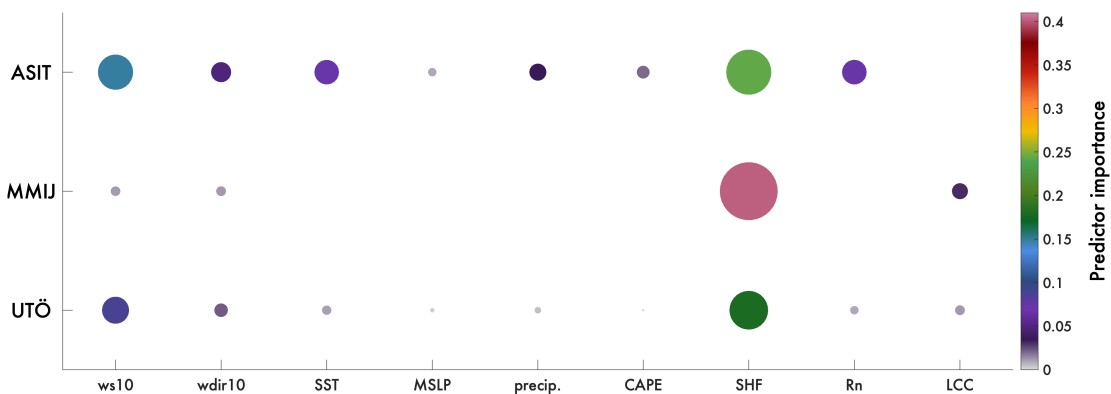

**Figure 8.** Predictor importance for single-level ERA5 variables at all three sites for the RF model in the binary (yes/no) LLJ classification task. Circles are both colored and scaled relative to their predictor importance.

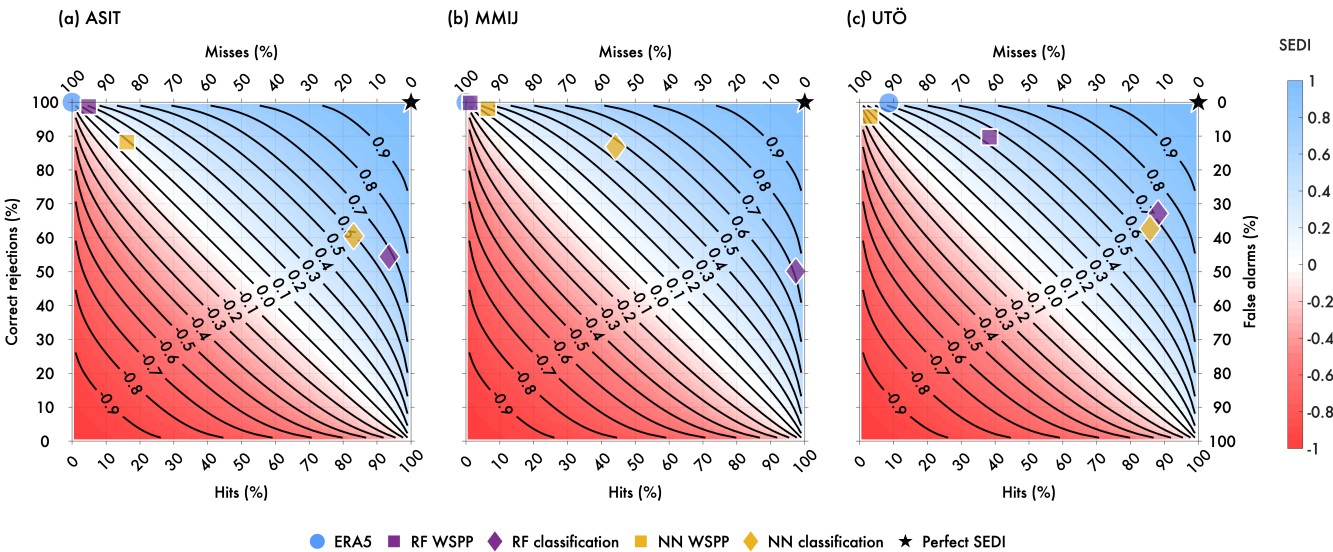

**Figure 9.** SEDI plots for (a) ASIT, (b) MMIJ and (c) Utö. On the lower abscissa, the percentage of hits is plotted, i.e., the hit rate, and on the upper abscissa the percentage of misses. On the left ordinate the percentage of correct rejections is plotted, and on the right ordinate the percentage of false alarms, i.e., the false alarm rate. The background coloring indicates the SEDI which is also shown by the isolines. What would be a perfect SEDI (100% hits, 0% false alarms) is marked by the black pentagon in the top right corner of each panel. Results for ERA5 (blue circles), RF (purple symbols), and NN (yellow symbols) are plotted.





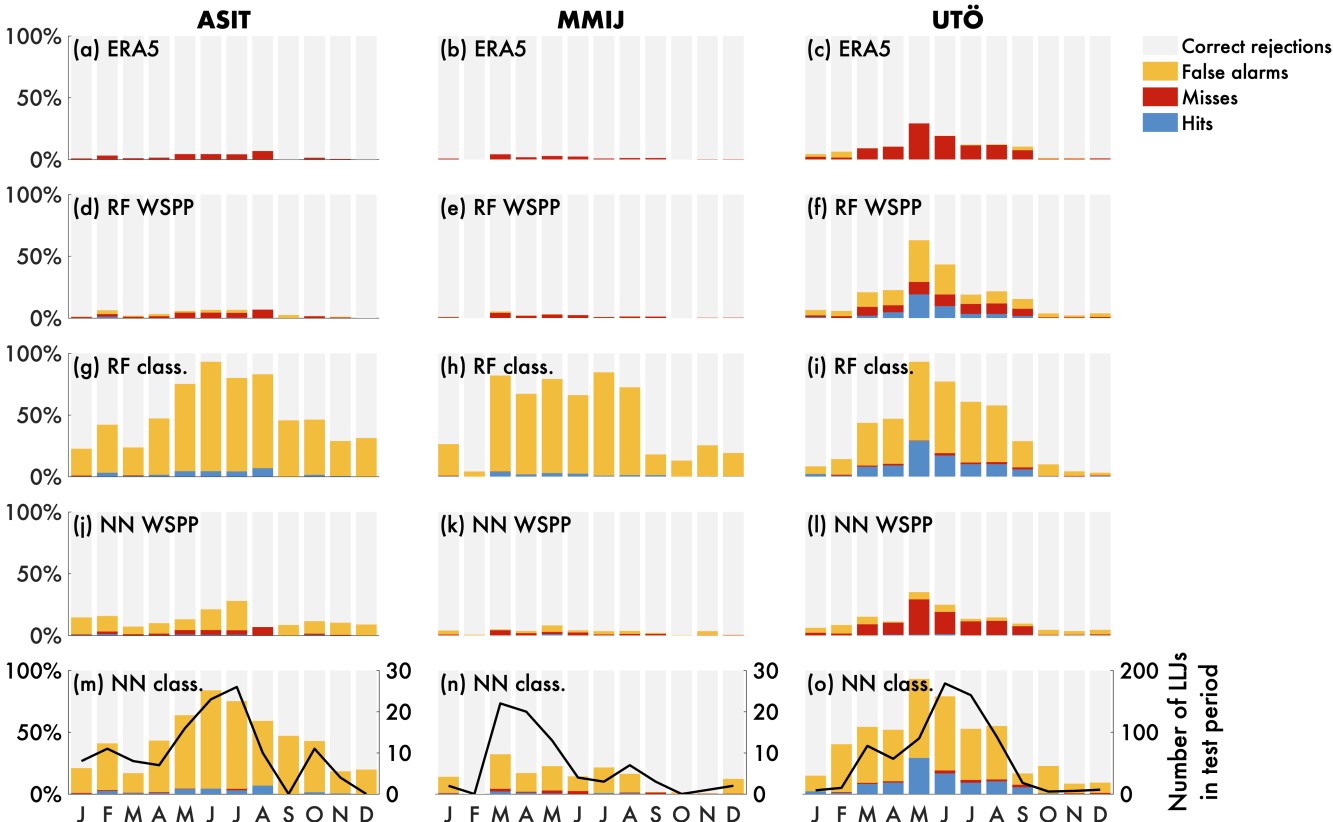

**Figure 10.** Seasonality of binary LLJ (crit. 1) predictions for (a–c) interpolated ERA5 wind speed profiles, (d–f) the RF WSPP regression task, (g–i) the RF classification task, (j–l) the NN WSPP regression task, and (m–o) the NN classification task. Seasonality is expressed as monthly proportions of correct rejections (light gray), false alarms (yellow), misses (red), and hits (blue), calculated relative to the ground truth lidar profiles for the test period. In panels m–o, also the total number of LLJs identified in each month in the test period is plotted.

The seasonality for LLJ predictions for the classification and WSPP tasks within the test period is evaluated in Fig. 10

in terms of the false alarms, misses, hits, and correct rejections, calculated relative to the ground truth lidar profiles. The seasonality is also plotted for the interpolated ERA5 profiles to compare and contrast ML model performance relative to ERA5 on a monthly basis. Generally, ERA5 is consistently missing LLJ occurrence across all months, with no hits at all (panels a–c). The ML models improve upon these results by increasing the monthly frequency of hits, particularly in the warmer months and particularly for Utö. For all sites, the classification models from both RF and NN correctly predict the majority of LLJs and the

RF WSPP also correctly predicts the majority of LLJs for Utö. However, all ML models introduce a high rate of false alarms, and this is particularly prominent for the classification models. This may be due to the fact that the models are optimized using the SEDI and that the RF is optimized using a cost matrix. Further refinement of the model optimization and development may reduce the high rate of false alarms.





## 4 Discussion

Generally, the ML models developed in this work offer an improvement over ERA5 in the regression task – predicting coastal wind speed profiles from single-level variables. In particular, both the RF and the NN generally result in lower RMSE than ERA5 at typical offshore hub heights when compared to the ground truth lidar profiles. Given these findings, it may be possible to utilize trained ML models to improve historical wind speed profiles from ERA5 (non-linear bias correction), as the ML models provides improved information on the wind speed profile based on ERA5 predictors, while the ERA5 wind profile

itself is not always very representative for local behaviour. Further, it is likely that similar ML models could be developed using direct field observations, i.e., from buoys and met masts, to be used for real-time wind profile predictions.

In particular, both the RF and the NN exhibit high fidelity when compared to ERA5 for reducing the RMSE for hours in which an LLJ is present. For the ML classification task of binary LLJ prediction, both ML models produce higher SEDI than ERA5, although both models suffers from a high rate of false alarms, an artefact of optimizing for the SEDI and a

consequence of the cost matrix for the RF method. Since correct rejections are numerous, it might be beneficial to optimize the ML methods for a measure not including those, such as e.g., the F1 score or the critical success index (CSI). Utilizing a larger set of predictors, the false alarm rate could be improved (however, see further discussion below why not more predictors were included in the study). Fig. 10 brings up many questions relevant for further investigation, such as (1) is there anything specific about LLJs that are missed or hit by the ML (e.g., hypothetically the more intense LLJs could be easier to predict), and (2) are

persistent LLJs more likely to be predicted?

Regardless of the statistical differences between the model architecture of the RF and NN models, both methods show high $PI$ for SHF and ws10. In at least one month at each site, all of the predictors demonstrate significantly different median values during LLJ hours when compared to non-LLJ hours as described by results from the Mann-Whitney U Tests (Fig. 4). However, the median SHF, which varies between negative and positive values depending on the month, exhibits significantly

higher values during LLJ hours, when the median SHF always is positive, for all three sites and across all months. In the WSPP regression task, the $PI$ is consistent across all vertical levels, indicating that dynamic readings of ws10 and SHF are highly important in reducing the RMSE of the predicted wind speed profiles. This high predictor importance is also observed for the classification task, indicating that ws10 and SHF maintain high $PI$ even when Monin-Obukhov similarity theory is invalid, i.e., due to the presence of an LLJ. The high predictor importance of SHF may be attributed to its relation to stability, particularly

for air-sea temperature differences.

Both the RF and the NN are sensitive to the quantity of training data available, with both methods having higher fidelity in both ML tasks (classification and regression) for Utö, the site for which the most lidar data is available for training. In particular, the NN exhibits higher sensitivity to a paucity of training data when compared to the RF. This discrepancy highlights the challenge in choosing adequate ML models for various applications; although the NN is traditionally more complex and

thus more capable of making more sophisticated statistical inferences, it is more sensitive to the body of training data and thus may produce weaker results than would be expected a priori (Wang et al., 2017). Initial testing, with a larger set of single-level variables from ERA5, show improved results for both models, both in terms of the RMSE and SEDIs, with the NN ahead of RF.



However, as high cross-correlation between the variables makes the analysis of predictor importance finicky, we decided to not include it. Although not applied in this study, spatial variation in the single-level ERA5 data could also be used as predictors for the ML models, making it possible to identify conditions on both the meso- and synoptic scale that favour LLJ formation, e.g., coastal effects and fronts.

It is of course also interesting to test whether the ML models created for Utö could used to predict the wind conditions at ASIT or MMIJ. While this research question lies beyond this study, it is possible to hypothesize that, as wind conditions at coastal sites are sensitive to the local conditions, e.g., if the wind is directed from the land or the sea, a model optimized for one of the sites is of relatively low value for the other sites. Switching from model output to locally performed measurements, these site specific characteristics of the ML models will be even more pronounced. However, in homogeneous surroundings, e.g., far offshore, it is likely that the models will be more general. The decrease in performance of the ML models with increasing distance to the origin of training data is suggested to be further analyzed.

Different optimization metrics could be introduced depending on the wind energy application under consideration, i.e., if a high number of false alarms would be detrimental, a different score that minimizes false alarms could be utilized. Further, given the improvement in results for Utö as compared to the other sites, it is possible that longer data sets in areas of higher LLJ frequency may be the best candidates for employing the ML methods presented here.

## 5 Summary and conclusions

Two ML tasks are investigated using multiple years of ERA5 reanalysis and lidar profiles at three offshore sites (the U.S. Northeastern Atlantic Coastal Zone, the North Sea, and the Baltic Sea) of relevance to wind energy. The regression task of accurately predicting the wind speed profile aloft, up to a maximum of 315 m a.s.l, is investigated through development of two ML methods, RF and NN, using nine selected single-level ERA5 variables as predictors and observed wind speed profiles by vertical scanning lidars as the ground truth. All selected single-level ERA5 variables are physics-based and chosen either for their relevance in describing synoptic and local scale atmospheric conditions or their ability to be measured using real-time observations within a wind farm, or both. The ws10 and the SHF have by far the highest $PI$ in WSPP. Further, these two variables are also the most important in predicting the LLJ and can help in assessing expected production from a farm and loads on a turbine. This indicates that both variables are highly important for describing local and mesoscale processes that influence wind speed profile development.

At heights below approximately 200 m a.s.l., both the RF and the NN have lower RMSE when compared to ERA5 for predicting wind speed profiles during hours in which an LLJ is present within the lidar profiles. Further, both models exhibit markedly higher SEDIs than ERA5 for binary low-level jet predictions using the single-level ERA5 variables as predictors.

Future work could employ the use of more training data when developing the models. This would be especially helpful for the NN, which demonstrates a high sensitivity to a relative paucity of training data for both ML tasks. However, the RF would also benefit from more training data, and thus consistent and accurate long term measurements with high vertical resolution are important. The results presented herein show promise for utilizing field measurements to make real-time predictions of coastal





wind speed profiles, particularly during conditions or at heights for which the Monin-Obukhov similarity theory is invalid, i.e., in case of LLJs or negative shear in general. Results also show promise for making wind speed profile predictions from models that may lack sufficient vertical resolution at heights relevant to wind energy.

**List of acronyms and abbreviations**

**ANN**   Artificial Neural Network

      **ASIT**   Air Sea Interaction Tower

      **CAPE**   Convective Available Potential Energy

      **CSI**   Critical Success Index

      **ECMWF**   European Centre for Medium-Range Weather Forecasts

**ERA5**   ECMWF Reanalysis version 5

      **IEA**   International Energy Agency

      **IFS**   Integrated Forecasting System

      **LCC**   Low Cloud Cover

      **LCoE**   Levelized Cost of Energy

**Lidar**   Light Detection And Ranging

      **LLJ**   Low-level jet

      **ML**   Machine Learning

      **MMIJ**   Meteorological Mast IJmuiden

      **MSLP**   Mean Sea Level Pressure

**NN**   Neural Network

      **NWP**   Numerical Weather Prediction

      **PCHIP**   Piece-wise Cubic Hermite Interpolating Polynomial

      **PI**   Predictor Importance

      **precip.**   Precipiation

**RF**   Random Forest



**RMSE** Root Mean Squared Error

**Rn** Net radiation at the surface

**SEDI** Symmetric Extremal Dependence Index

**SHF** Surface Sensible Heat Flux

**SST** Sea Surface Temperature

**wdir10** Wind direction at 10 m a.g.l

**WHOI** Woods Hole Oceanographic Institute

**WRF** Weather Research and Forecasting

**ws10** Wind speed at 10 m a.g.l

**WSPP** Wind speed profile prediction

*Code and data availability.* The code used to generate the figures can be acquired by contacting the corresponding author. For the ERA5 data, hourly values on model levels for wind components, temperature, and specific humidity, hourly data on a single-level for surface pressure, we refer to Hersbach et al. (2017). Data were downloaded from the Copernicus Climate Change Service (C3S) (2023). The results contain modified Copernicus Climate Change Service information. Neither the European Commission nor ECMWF is responsible for any

use that may be made of the Copernicus information or data it contains.

*Author contributions.* The conceptualization, administration, methodology, programming, validation, formal analysis, visualization and writing the original draft was performed by CH and JAA. CH was supervised by ES, SI and HK. JAA was supervised by RJB and SCP. Funding acquisition was carried out by ES, SI, HK, RJB and SCP. Data curation was performed by VV who also wrote Sect. 2.2.3 about the Utö lidar measurements. All authors participated in reviewing and editing the manuscript.

*Competing interests.* At least one of the (co-)authors is a member of the editorial board of Wind Energy Science. Apart from that, the authors declare no conflict of interest. The funding agencies had no role in the design of the study; in the collection, analyses, or interpretation of data; in the writing of the manuscript, or in the decision to publish the results.

*Financial support.* This research was funded by the Energimyndigheten (Swedish Energy Agency) VindEl program, Grant Number 47054-1. The work forms part of the Swedish strategic research program StandUp for Wind.





*Acknowledgements.*    ASIT: Thanks to Anthony Kirincich and Eve Cinquino for providing the ASIT data (doi: 10.26025/1912/27014). MMIJ: The data were collected by TNO as part of the research program *"Meteorologisch onderzoek windcondities op zee"* which run under FLOW (Far and Large Offshore Wind) supported by the Dutch Ministry of Economic Affairs and Climate Policy. Thanks to J.P. (Hans) Verhoef for your help accessing the MMIJ lidar data.



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
