# Peer review of "Machine Learning Methods to Improve Spatial Predictions of Coastal Wind Speed Profiles and Low-Level Jets using Single-Level ERA5 Data"

_Wind Energy Science, 2023_

## Author Comment (AC1)

**Author's response**

Dear Editor and Dear Referee #1,

Thank you very much for your feedback that helped us further improve and strengthen our manuscript. An overview of the changes in the manuscript related to your comments are presented below. However, as one of the main authors of the manuscript (J.A. Aird), who was in charge of the coding and implementation of the neural network (NN), has moved on to other work outside of research, our possibilities to re-run the machine learning (ML) models are very limited. In the following, we have performed the suggested tests for the random forest (RF) model and argue that the results should also hold for the NN model. We hope you will have indulgence with this and that you still find our presented results worth publishing.

**Specific comments and questions**

**P1L17: The Electricity generation number is the capacity, I presume. This should be spelled out.**

Lines 17–18 now reads: *[...] each year there is an increase in the installed capacity of wind power by approximately 136 billion kWh (EIA, 2023).*

**P8L186: Were the chosen ERA5 grid points the nearest water point? or were land points used for any of the sites?**

No grid points on land were used. We have added the value from the ERA5 land/sea-mask in the manuscript, and lines 193–194 now reads:

*The values of the ERA5 land/sea-mask (ranging from 0 for open sea to 1 for land) for the selected grid points were 0.04 for ASIT, 0 for MMIJ, and 0.01 for Utö.*

**P11L260: I assume the standardization mentioned here is subtracting the mean and dividing by the standard deviation to get z-scores. Please clarify.**

Yes, you are absolutely right. We have clarified in the manuscript that it is the z-score normalization that we have used.

**P21L407: You write "In particular, both the RF and the NN generally result in lower RMSE than ERA5 at typical offshore hub heights when compared to the ground truth lidar profiles.", but that does not seem true to me at Utö. Looking at Fig. 6 it seems to me the RF model is on par with ERA5 at relevant hub heights, and NN is worse than ERA5. The same can be set for the NN method at ASIT.**

Thank you for noticing this. We have changed the text in the Discussion which now reads (line 425–428):

*For both ASIT and MMIJ, the RF and the NN result in lower RMSE than ERA5 in the lower half of the height range swept by the turbine blades (hub height and down) of a typical modern offshore wind turbine, when compared to the ground truth lidar profiles (Fig. 6). For Utö, the performance of the ML models is similar to that of ERA5.*

We have also re-plotted Fig. 6 in the manuscript to present the results in a clearer way, please see Fig. R1 below.

[Figure]

Figure R1: (a–c) RMSE for wind speed profiles for ERA5 (gray), the RF (purple), and the NN (yellow) for ASIT, MMIJ, and Utö. The values of RMSE are calculated relative to the lidar observations and the results are valid for the test period. Panels (d–f), present RMSE for ERA5 and the ML models for the subset of the test data with LLJs (crit. 1) in the lidar profiles.

**You did a good job of splitting the data into training, testing, and validation data and taking care of the data stratification (seasonality) and correlation (excluding data between sets). However, you could do more to investigate the sensitivity of the results to the splits, e.g. you could have done a k-fold cross-validation of the training+testing data (adjusting exclusion data each time) and presented the robustness across folds. This would also give you a better sense of the robustness of your predictor importance analysis.**

The robustness of the predictor importance is a very important remark. We have tested the sensitivity of the results on the predictor importance by re-running the RF for the WSPE regression task, testing 10 randomly created splits of the time series for each of the three sites. The first split of the time series (index 1) is the same that was used for the results presented in the manuscript. In Fig. R2, the predictor importance for all predictors at the measurement height closest to 150 m (140 m for ASIT and MMIJ, 152 m for Utö), which is a typical hub height for modern offshore wind turbines, is presented. Results are for the validation period just as in Fig. 5 in the manuscript, but look similar also for the test period.

[Figure]

Figure R2: Predictor importance for 10 random splits of the time series as calculated by the RF for the measurement height 140 m (ASIT, MMIJ) or 152 m (Utö). The results are for the validation period.

As seen in Fig. R2, there is some variation in the predictor importance for the different predictors across folds. However, the general conclusions remain that, for all sites, the ws10 is by far the most important predictor for the wind speed higher up, followed by the SHF. For ASIT, the wdir10 also has slightly higher predictor importance than the rest of the predictors, while for MMIJ and Utö it is of similar (un)importance.

Lines 359–363 in the revised manuscript now reads:

*Further, to test the robustness of the method and the sensitivity to the data split (Fig. 3) the time series have been randomly split into training, validation, and testing ten times for all sites. Then the RF has then been trained according to the protocol described in Sect. 2.5 and the predictor importance calculated. Results from this sensitivity analysis (not shown) clearly separates the ws10 as the most important predictor across all folds and for all sites, and SHF as the second most important predictor.*

**In the same spirit as the question above, it would also be interesting to know more about how testing and validation results differed.**

In Fig. R3, we have plotted the RMSE difference (RF minus ERA5) for the WSPE task for the ten folds of the time series previously described. Negative numbers implies that the RF performs better than ERA5 (lower RMSE), and vice versa for positive numbers. We see that, for all three sites, the results for the test period and the validation period generally resembles each other, although there is of course some variation comparing different folds. For MMIJ the results for the validation period are generally associated with greater improvements than the results for the test period, but for ASIT and Utö differences are smaller and the performance in the test and validation periods more similar. The thick lines in the plots, marking the results using time split number 1 for the test period (which are the results presented in Fig. 6 in the manuscript) are more-or-less close to the middle of the distribution of the results for the test period for MMIJ and Utö, but slightly shifted to the right (lower performance of the RF) for ASIT. Thus we claim that the results presented in the manuscript are representative for the method, or at least not indicating results that are better than the average.

[Figure]

Figure R3: Comparison of RMSE differences (RF minus ERA5) in the WSPE regression task for the test and the validation period for all ten random data splits.

Although not tested for the NN – for reasons described above – we believe that the conclusions regarding the predictor importance and the RMSE performance, across the folds are general.

Lines 380–382 in the revised manuscript now reads:

*In the same sensitivity test described above, running the RF for ten random splits of the time series at each site individually, the RMSE profiles are similar to what is presented here, and for ASIT even showing slightly larger reductions on average.*

**You evaluate the wind speed profile with the RMSE. The same metric your ML models are optimized for. For the methodology to be truly and generally valuable, I would consider it important to see improvements or at least no major degradation, for other statistics as well. For example, you spend some time in the introduction emphasizing the importance of correctly modeling the shear affecting the rotor. Why not show how well the shear is estimated by your ML approach? A second thing you could also consider is how well distributions are modeled. These ML methods are good mean-finding methods for different conditions. This may give you a low RMSE, however, this can come at the cost of narrower distributions that greatly underestimate the average power density.**

Thank you for sharing this concern. We are aware of this problem and agree that one should be very careful assuming that the output from ML models, not only performs better for the metric it was optimised for, but also in every other aspects. Our only goal with the WSPE regression task is to improve the RMSE for wind speed profiles. As the ML models are optimised using all wind profiles in the validation period, it was expected that there should be an improvement for the general wind profile in the test period, but not necessarily so if an LLJ (which to some extent can represent an extreme case of a wind speed profile) was present. We strongly suggest to optimise the ML models for the main topic of interest. If "everything" is of interest, then multiple verification metrics could be included in evaluation and optimization of the models. In Fig. R4, the wind speed distribution at the height level in the data closest to 150 m is plotted for the lidar, ERA5, RF, and NN for the three sites. As seen in the figure, ERA5 is better at representing the full distribution, especially at ASIT and Utö, while the ML models tend to overestimate the peak of the distribution. With your comment and this figure in mind, we have added the following sentences in the revised version of the manuscript (lines 432–438):

*However, it should also be mentioned that – as the ML methods applied here focus on reducing the RMSE – there is a possibility that the quality of the prediction is inferior in other aspects when compared to ERA5. For example, analysing the wind speed distributions at 150 m height for the three sites (not shown), ERA5 better represents the full distributions while both RF and NN tend to underestimate the extremes and overestimate the peak of the distributions. This could have further implications, e.g., in non-linear calculations such as in estimations of the power production. We strongly recommend to optimise the ML models using the metric (or combination of metrics) that are of importance for the application, and to be aware that an improvement for one metric does not necessarily result in an improvement for all other metrics.*

[Figure]

Figure R4: Wind speed distribution at the height level closest to 150 m. The results presented are for the test period.

**How did you treat/encode the circularity of wdir10 in the ML models? it is not clear to me if you used any kind of encoding, e.g. sin and cos transformation**

You are right, this was not clearly written in the manuscript. We used 0–360° for the wind direction, which we realise now is not the best way to do it. For reasons explained above, we are unfortunately not able to retrain the NN using the sine and cosine transformations of the wind direction. However, as a test, we have implemented this for the RF and recalculated the predictor importance for the binary prediction of the LLJ. See further information below as a reply to your next comment.

**You state in your introduction that an important mechanism for LLJ-genesis is warm air advected over cold water so I would assume wind direction to be of some importance (given that the sites are coastal). However, wdir10 appears to be of minor importance (perhaps a bit more important at ASIT). I think it would be good to explain or at least discuss this more. I'm wondering if it's connected to the question of encoding of circularity and the fact that at least ASIT and Utö clearly have land to the North. In both cases, encoding wind direction from 0° to 360° clockwise from North would make it difficult, e.g. for the decision trees, to split land directions from sea directions (since land directions are at both tail-ends).**

Thank you for noticing this. Coming back to what we explained in our reply to the previous question, the results for the predictor importance has been recalculated with the RF using the sine and cosine encoding and results are presented in Fig. R5. For comparison, also the results achieved when using 0–360° for the wind direction are included (panel a). As seen in the figure, introducing the sine and cosine transformation of the wind direction changes which predictors that are selected in training the RF. For example, for ASIT, all predictors (except LCC) are included if wdir10 is used, but only ws10, SHF, and Rn are included if sin(wdir10) and cos(wdir10) are used.

Concluding from Fig. R5, the statement in the manuscript that SHF is the most important predictor in LLJ prediction still holds, and also that it is followed by ws10. It also seems that the new encoding of the wind direction does not help to dramatically increase the predictor importance of the wind direction. The relatively low importance of the wind direction is interesting, because – as you mention – the warm air advection over relatively cold water in spring/early summer is one of the main explanations for LLJ formation. However, it is important to note that warm air advection over cold water results in stable stratification and a change in SHF. Since the same wind direction not necessarily result in stable stratification and good conditions for LLJ formation in other seasons, the SHF is a much better predictor than the wind direction.

[Figure]

[Figure]

Figure R5: Predictor importance from the RF model in binary LLJ prediction. Results are for the validation period. For the results in panel (a) the wdir10 is treated as values in the range 0–360°, while for the results in panel (b) the wdir10 is encoded as sin(wdir10) and cos(wdir10).

In the manuscript, we have added the following to the Discussion (lines 457–464):

*Note that, throughout this study, the wind direction (wdir10) was included as values in the range 0–360°. Using the sine and cosine encoding of the wind direction to better handle the circular coordinates of the variable was tested for the analysis of predictor importance in binary predictions of the LLJ using the RF method, and although this representation causes minor differences in predictor importance, the main*

*conclusion that SHF and ws10 are the most important predictors still holds. It is also worth noting that, although warm air advection over relatively cold water in spring and early summer, creating stable conditions, is often mentioned as one of the key formation mechanisms for offshore/coastal LLJs, it is the stability of the atmosphere rather than the wind direction that is the win direction itself that is most important, i.e., the SHF is more important than wdir10 (no matter the encoding) for LLJ prediction.*

**Technical corrections**

Thank you for all these technical corrections. If nothing else is stated, we have changed according to the suggestions.

**P6L143: "compare" should be "compared"**

**P8L196: I assume you mean correlation greater than 0.5, so shouldn't it be ">"' here?**

Here we actually mean that we have selected variables with a low cross-correlation. The reason for this is to get a much more straight-forward analysis of the predictor importance. We have clarified this further in the manuscript (lines 202–207):

*Based on the long list of single-level variables provided by ERA5, a selection of nine variables, all with Pearson correlation coefficients in the range -0.5 to +0.5, was performed. [...] The low cross-correlation assures a high degree of independency among variables which in turn simplifies the analysis of how important the individual variables are in terms predicting the wind speed profile.*

**P9: Predictor importance (PI) should not be italicized**

**P12L305: SEDI should not be italicized here.**

**P17, Fig. 6 caption: It says ERA5 (dotted), but I see no dots.**

Once again, thank you for all your comments.

Sincerely,
C. Hallgren, J.A. Aird and co-authors

**References**

EIA. U.S. Energy Information Administration, 2023. URL https://www.eia.gov/international/data/world. last access: 2023-07-02.

---

## Author Comment (AC2)

**Author's response**

Dear Referee #2,

Thank you for your very positive review of our manuscript. Replies to your specific comments are presented below.

**Summary:**

**This paper investigates the development of machine learning (ML) models to predict coastal wind speed profiles and LLJ occurrence from single-level meteorological variables Data from three locations of high relevance to offshore wind energy deployment (the U.S. Northeastern Atlantic Coastal Zone, the North Sea, and the Baltic Sea) were used. The ML models are trained on multiple years of lidar profiles and utilize single-level ERA5 variables as input. The models provide output spatial predictions of coastal wind speed profiles and LLJ occurrence.**

**General Comment:**

**The study is interesting and valuable for the offshore wind energy community. The article is well-written. The authors have used a variety of locations with different wind characteristics to apply the methods, which increases the applicability of the study.**

**Specific comments:**

**ERA5 data is quite coarse for wind applications in coastal areas. Could the authors comment/discuss if there is a potential advantage of using wind data of higher resolution for both LLJs and coastal wind speed profiles?**

This is a fair point. In a paper by Hallgren et al. (2020) the performance of ERA5 was compared to other reanalyses/wind atlas of both higher and coarser resolution at four sites in the coastal waters of the Baltic Sea where lidar observations were available (Utö was one of the sites in the study). It was concluded that, in terms of the average wind profile, ERA5 (approximately 17 km × 31 km grid resolution in the Baltic Sea) demonstrated similar error metrics as the regional reanalyses UERRA (Uncertainties in Ensembles of Regional Reanalyses, 11 km × 11 km grid resolution) and as the New European Wind Atlas (NEWA, 3 km × 3 km grid resolution). However, the other global reanalysis that was investigated, MERRA2 (Second Modern-Era Retrospective analysis for Research and Applications) with coarser resolution (40 km × 55 km) did not achieve comparable results, heavily underestimating the average wind speed in the profile at all sites.

In terms of LLJs, the same study concluded that UERRA was the model that best captured the frequency of LLJ occurrence and its seasonal variation. All reanalyses struggled resolving LLJs at the correct point in time, with hit rates of 12–18% for ERA5, 28–41% for UERRA, and 15–25% for NEWA for the four sites. False alarm rates were on the other hand relatively low, indicating a general underestimation of LLJs; 1–3% for ERA5, 4–7% for UERRA, and 1–4% for NEWA. Recalling from Eq. 5 in the manuscript, the hit rate and the false alarm rate combines to give the SEDI.

In similar work by Kalverla et al. (2020), ERA5 was compared to both to NEWA and to the Dutch Offshore Wind Atlas (DOWA) with 2.5 km horizontal resolution. The study focused on lidar measurements performed at MMIJ in the North Sea. The authors conclude that DOWA describes the average wind profile best in general, however, with ERA5 outperforming the other models in strongly stable conditions. The relative occurrence of LLJs is too low in all models compared to observations and the hit rate of perfectly timing the occurrence of LLJs was 28% in ERA5, 52% in DOWA, and 33% in NEWA. False alarm rates were low for all models; 0.2% for ERA5, 1% for DOWA, and 0.6% for NEWA. Note that a slightly different definition of the LLJ was used by Kalverla et al. (2020) as compared to Hallgren et al. (2020).

In a recent preprint, Sheridan et al. (2023) analysed model performance comparing model output with lidar measurements performed at two sites off the California coast and focusing on LLJs. ERA5 was compared to two regional models with different planetary boundary layer (PBL) schemes, the National Renewable Energy Laboratory (NREL) data set for the Outer Continental Shelf off the coast of California (CA20-Ext) and the

2023 National Offshore Wind data set (NOW-23), both with 2 km horizontal resolution. Using a high threshold for LLJ identification, ERA5 failed to accurately resolve any LLJs at the correct point in time, and also false alarms rates were very low, 0.05–0.08%. CA20-Ext had a hit rate (false alarm rate) of 47–52% (2–4%) and for NOW-23 the scores were 13–23% (9–10%). Unfortunately for our work, there is not yet any similar work for comparison off the US Atlantic coast.

Concluding from these three studies, it seems that – in terms of the average wind conditions – ERA5 performs reasonably well, also when comparing with higher resolution models. However, in terms of resolving the LLJs, ERA5 struggles in getting the relative frequency of occurrence correct, and as a consequence of this also with accurately predicting the presence of an LLJ in time.

In the revised version of the manuscript, one of the paragraphs in the Introduction now reads (lines 83–91):

*Although LLJs have been observed and simulated frequently offshore at heights relevant to wind energy, numerical weather prediction (NWP) models exhibit difficulty in resolving LLJ characteristics with high accuracy, i.e., in terms of timing and morphology (jet core height and speed) of LLJs. Kalverla et al. (2020), Hallgren et al. (2020), and Sheridan et al. (2023) all showed that regional models, optimised for a specific region and with higher horizontal resolution than the global models, are better in resolving coastal LLJs in the North Sea, the Baltic Sea, and off the California coast, respectively. However, not only the horizontal resolution is crucial, but also how calculations in the boundary-layer are treated by the models, i.e., the PBL scheme. When it comes to the average wind conditions in the profile, it seems to be less of a difference between state-of-the-art models, as long as the horizontal and vertical resolutions are good enough (Hallgren et al., 2020), even if the model performance varies with atmospheric stability (Kalverla et al., 2020).*

**It would be great that the authors provide some numbers regarding the computational time/power used by the different methods.**

Unfortunately, as one of the main authors of the manuscript (J.A. Aird), who was in charge of the coding and implementation of the neural network (NN), has moved on to other work outside of research, our possibilities to re-run the NN models are very limited. Here we only present results for the random forest (RF) for the different tasks, but run-time for the NN should be of comparable numbers. We hope you will have indulgence with this.

The time it takes to train the RF, i.e., finding the optimal set of predictors using the forward and backward selection methods, is presented in the Table below. How long it takes to reach the optimal set of predictor mainly depends on how many predictors are chosen by the algorithm, and since this varies between height levels the run-time is presented as a min-max interval for each site. The run-time also depends on the amount of training data, which mainly differs between the sites, but also the data availability differs between different height levels for each site. The task is perfectly parallelizable as the search for the best predictors on one height level is completely independent of the results on other height levels.

For the binary task of predicting LLJs, the training time also depends on the time to find the optimal cost matrix, on top of the time it takes to find the best set of predictors using the forward and backward selection methods. For the LLJ prediction task, the ML models only need to be run once. All run-times presented in the Table below are representative for when the RF is run on a standard laptop (MacBook from 2022, 8 GB RAM, 8 cores).

|  | ASIT | MMIJ | Utö |
| --- | --- | --- | --- |
| Wind speed, one height level | 286 – 308 s | 243 – 338 s | 207 – 486 s |
| LLJ prediction | 164 s | 145 s | 388 s |

The time to calculate predictor importance is not included in the numbers presented in the Table.

In the revised manuscript, the following has been added (lines 352–354) for the WSPE task:

*The run-time to find the optimal set of predictors varies between height levels and among the sites, but generally ranges from 4 to 8 minutes per height level on a standard laptop.*

For the LLJ prediction, the following was added on lines 396–397:

*Finding the optimal predictors for the LLJ classification tasks takes, depending on the site, between 2.5 and 6.5 minutes on a standard laptop.*

Once again, thank you for your comments.

Sincerely,
C. Hallgren, J.A. Aird and co-authors

**References**

Christoffer Hallgren, Johan Arnqvist, Stefan Ivanell, Heiner Körnich, Ville Vakkari, and Erik Sahlée. Looking for an Offshore Low-Level Jet Champion among Recent Reanalyses: A Tight Race over the Baltic Sea. *Energies*, 13(14):3670, 2020. doi: 10.3390/en13143670.

Peter C Kalverla, Albert A M Holtslag, Reinder J Ronda, and Gert-Jan Steeneveld. Quality of wind characteristics in recent wind atlases over the North Sea. *Quarterly Journal of the Royal Meteorological Society*, 146(728):1498–1515, 2020. doi: 10.1002/qj.3748.

Lindsay M Sheridan, Raghavendra Krishnamurthy, William I Gustafson Jr, Ye Liu, Brian J Gaudet, Nicola Bodini, Rob K Newsom, and Mikhail Pekour. Offshore low-level jet observations and model representation using lidar buoy data off the California coast. *Wind Energy Science Discussions*, 2023:1–28, 2023. doi: 10.5194/wes-2023-152.